# Political and personal reactions to COVID-19 during initial weeks of social distancing in the United States

**Sarah R. Christensen, Emily B. Pilling, J. B. Eyring, Grace Dickerson, Chantel D. Sloan, Brianna M. Magnusson***

Department of Public Health, College of Life Sciences, Brigham Young University, Provo, Utah, United States of America

\* Brianna_Magnusson@byu.edu

## Abstract

### Objective

To examine perceptions, behaviors, and impacts surrounding COVID-19 early in the pandemic response.

### Materials and methods

A cross-sectional survey of 1,030 U.S. adults was administered on March 31st, 2020. This survey examined attitudes toward media, government, and community responses to COVID-19 by political ideology and sociodemographic factors. Knowledge, anxieties, and impacts of COVID-19 were also assessed.

### Results

Conservatives were more likely to report that COVID-19 was receiving too much media coverage and people were generally overreacting; liberals were more likely to report the government had not done enough in response to the pandemic. Females and those with lower income experienced more COVID-19 related economic anxieties. Those working and with children at home reported higher social, home, and work disruption. Social distancing behaviors were more common among liberals and were associated with increases in depressive symptoms. General knowledge about COVID-19 was widely exhibited across the sample, however, Black and Hispanic respondents were less likely to correctly answer questions about the availability of a vaccine and modes of transmission.

### Conclusions

Public health experts should consider the political climate in crafting messaging that appeals to the values of those across the political spectrum. Research on the COVID-19 pandemic should continue to monitor the effects of social distancing on mental health and among vulnerable populations.

**Data Availability Statement:** We have published our data as well as the accompanying codebook and survey on OpenICPSR. https://doi.org/10.3886/E119629V1.

**Funding:** The authors received no specific funding for this work.

**Competing interests:** The authors have declared that no competing interests exist.

## Introduction

The COVID-19 pandemic of 2019–2020 presents physical, social, emotional, and financial challenges. Effective communication of public health information is central to a successful pandemic response. The first confirmed case of community transmitted COVID-19 in the U.S. was reported on February 26th, 2020. By March 31st, cases rapidly increased to over 181,000 –the highest reported incidence of any country. By August 20th, 2020, confirmed cases in the US reached over 5.5 million and deaths due to COVID-19 exceeded 172,000. [1] National and state responses to the crisis quickly became the central messaging for all major news outlets. Local governments implemented varying restrictions and strategies to interrupt outbreaks and mitigate disease burden in their communities. Concern exists that variation in messaging across communities is resulting in inconsistent expectations, anxieties, and responses among the public.

Recent studies conducted in China and Italy have investigated the psychological and mental health impacts of the COVID-19 pandemic on patients, health care workers, and the general population. [2–6] China's general population experienced an increase in panic disorder, depression, and anxiety after strict quarantine measures were enforced. [3] Over one-half of China's general population reported moderate-to-severe psychological impacts and one-third reported increased levels of anxiety. [6] Groups in China's general population that were associated with worse mental health outcomes included females, students, those with underlying health conditions, and those with specific symptoms. Among health care workers in China, there was an increase in depression, anxiety, insomnia, and distress. [2] Women, nurses, those in Wuhan, and front-line health care workers were all groups that were associated with worse mental health outcomes. [2] Similar results were found in health care workers in Italy. [4] About half of the health care workers sampled showed signs of PTSD, about one-fourth showed signs of depression, about one-fifth showed signs of anxiety and stress, and 8% showed signs of insomnia. Groups that were associated with worse mental health outcomes included those who were younger, female, and front-line health care workers. [4] Finally, a systematic review of studies relating to the mental health impacts of the COVID-19 pandemic broadly confirmed many of these more specific results in showing that the most common adverse psychological impacts across the general, health care, and clinical populations were PTSD, depression, and anxiety. [5] Populations that were especially vulnerable to worse mental health outcomes included health care workers and patients with severe complications, and groups that were associated with worse mental health outcomes included females and young people.

Our study further analyzes this phenomenon in the general U.S. population and focuses on bringing further understanding to the differential effect of the COVID-19 pandemic on subsets of the general population.

Access to conflicting and misleading information in the media, as well as decreasing trust of government and scientific communities, impacts the ability to effectively communicate during crisis situations. [7] Media messaging holds immense power in swaying public trust in crisis response tactics. Research on media following Hurricane Katrina suggested that as media politicized coverage of disaster relief efforts, individuals faced extreme difficulty in forming independent opinions of crisis management processes. [8] As public opinion is swayed to align with polarized agendas, the ability of authority figures to promote coordinated responses with public support is severely undermined. [7]

Acknowledging underlying attitudes is crucial during the process of analyzing, predicting, and attempting to guide public behavior during a crisis. Consistent with Theory of Planned Behavior, [9] understanding the public's attitudes toward COVID-19 and the response of the media, government, and public to the pandemic may provide insight into behavioral intention

and adherence to preventative behaviors in the wake of this ongoing global pandemic. In light of differences in vulnerability to COVID-19, access to information, sources of information, and timing of national, state, and local responses to COVID-19, it is reasonable to assume that individual responses in the midst of the pandemic vary widely. This study examines individual attitudes, behaviors, anxieties, mental health impacts, and knowledge early in the pandemic response, as well as those outcomes by sociodemographic characteristics and political ideology. The purpose of this study was to determine how complex factors within society shaped early perceptions of and responses to COVID-19.

## Materials and methods

An anonymous cross-sectional internet survey was administered to 1,030 adults residing in the U.S. on March 31st, 2020. The sample was recruited by Qualtrics (Provo, UT, USA). Quotas for sex, race, and income, derived from U.S. Census data, [10] were used to increase demographic representation. Implied consent was provided prior to the survey. Participants were presented with information about the survey as well as potential risks, benefits and compensation. Following this information participants were presented with a statement which read, "The completion of this survey implies your consent to participate. If you choose to participate, please proceed with the questions." Compensation was valued at <$5 U.S. The study was approved by the Brigham Young University Institutional Review Board.

Questions assessed political ideology, scientific trust, and media consumption, as well as attitudes, anxieties, impacts, and knowledge related to COVID-19. Respondents also assessed mental health and demographic information.

Three author-constructed questions assessed attitudes toward the response to COVID-19. Respondents were asked about pandemic media coverage (too much, right amount, or too little), government action (not enough, responding correctly, or too much), and public response (overreacting, responding correctly, or too much). Respondents also answered two true/false questions: if they believed their state would experience a major outbreak of the virus and if they would isolate if they contracted the virus.

We used the Political Polarization in the American Public Survey, [11] which has been validated as a reliable measure of political partisanship, [12] to examine personal political ideologies through a series of 10 dichotomous statements on political issues. Responses were scored (liberal = -1 vs. conservative = +1), summed, and categorized as leans liberal, moderate, and leans conservative. Respondents also self-characterized their political views on a 7-point scale (extremely liberal to extremely conservative).

Two additional items asked about attitudes toward global warming (most scientists think global warming is happening vs. there is a lot of disagreement as to whether global warming is happening) [13] and trust in government to make vaccination decisions (I trust that the government makes the best decisions when it comes to vaccination requirements vs. I do not trust the government to make decisions about what vaccinations are required). [14] A subset of questions from the Reuters Institute Digital News Report [15] was used to assess media consumption. Respondents indicated their usual sources of media (ABC, FOX, NPR etc.) which were each given a bias score based on Ad Fontes Media's source evaluations. [16]

Pandemic-related behavior change was assessed by eleven author-constructed items asking respondents to compare their behavior on March 31st with their behavior before the pandemic on a 5-point scale from "much less than usual" to "much more than usual". Behaviors included virtual communication, face-to-face contact, visiting restaurants/bars, stores, work, and travel. Four items asked respondents to indicate agreement on a 7-point scale that "events related to

COVID-19 had interrupted" their social life, home life, work or vocational life, and/or hurt their mental health.

Fourteen author constructed items assessed pandemic-related anxieties using a 7-point agreement scale. Four statements related to fear if they, an older family member, a young family member, or a healthy adult family member became ill with COVID-19. Two statements assessed anxieties related to healthcare equipment and personnel, three statements assessed economic concerns, and four items assessed concerns related to children at home (e.g. routines affected, children without care, etc.). Questions regarding work- and child-related anxieties were only asked of those working for pay before the pandemic and those who had children under 18 in the household respectively. The last question assessed concerns that mental health would suffer due to social distancing measures.

Respondents assessed change in their own mental health from before the COVID-19 pandemic on a 7-point scale from much worse to much better. Respondents completed the PHQ-9 (a valid measure of depressive severity) [17] retrospectively for the two-week period preceding social distancing and for the current two-week period. Higher PHQ-9 scores indicated more depressive symptoms. An increase in depressive symptoms is indicated by a positive change score.

True/false and multiple-choice items developed by the authors were used to assess respondent knowledge regarding common symptoms of COVID-19, recommended preventative measures, viral spread, and comparisons of COVID-19 and seasonal influenza.

Respondents reported age, biological sex, race, ethnicity, marital status, education level, whether they were currently in school, employment status prior to the pandemic, average hours worked currently and prior to the pandemic, household size, household income, whether they received government nutritional program assistance, children at home, state of residence, and flu vaccination history.

Frequencies, proportions, and means were calculated. Chi-square, t, and F tests were used to examine the influence of demographic characteristics, political ideology, and mental health on attitudes, knowledge, anxieties, behavior change, and impact variables. Logistic regression was used to assess the relationship between political ideology and attitudes towards media, government, and community responses to COVID-19 while controlling for sociodemographic characteristics, political ideology, media bias, global warming agreement, and trust in government vaccination requirements. Initial covariate selection included all variables that were significant ($p < 0.05$) in bivariate tests, including: political ideology, bias score for consumed news media, attitudes toward global warming and vaccination, sex, race, poverty level, and education. The final model was achieved by sequentially removing non-significant predictors and assessing the impact on model fit using the Bayesian Information Criterion and Akaike Information Criterion. Non-significant predictors were retained if removing them worsened model fit.

Two attitude questions were dichotomized for logistic regression, due to small cell counts. Those who responded there was "too little media coverage" (n = 88) were merged with those who responded "the right amount" of media coverage. Similarly, those who responded the government had "done too much" (n = 51) were merged with those who responded the government had "done the right amount" in response to the pandemic. Logistic regression models were tested with the grouped categories as outlined above, and with the small categories coded as missing; results were similar.

Logistic regression was also used to assess the relationship between knowledge about COVID-19 while controlling for sociodemographic characteristics and media bias. Initial covariate selection included all variables that were significant ($p < 0.05$) in bivariate tests, including: bias score for consumed news media, sex, race, poverty level, and education. The

final model was achieved by sequentially removing non-significant predictors and assessing the impact on model fit using the Bayesian Information Criterion and Akaike Information Criterion. Non-significant predictors were retained if removing them worsened model fit. All analyses were completed in SAS 9.4.

## Results

The sample included 1,030 U.S. adults from 48 U.S. states and D.C. No respondents resided in Vermont or Wyoming. The sample was 47.5% male, 49.0% white, 23.4% Black, 12.0% Hispanic and 12.3% Asian. About 29% of the sample had children under 18 living at home, about 23% were living under the Federal Poverty Line, and about 24% had received government benefits in the last six months. Sample demographics overall and stratified by political ideology are presented in Table 1.

The majority (63.1%) felt that COVID-19 was receiving the right amount of media coverage, with 28.4% responding that the pandemic was receiving too much media coverage. Conservatives were most likely to feel the pandemic was receiving too much media coverage (50%) compared to moderates (30.1%) and liberals (21.0%; p-value <0.001). Table 2 provides the adjusted logistic regression analysis for the attitude questions. Compared to conservatives, liberals had three times the odds (aOR 3.3; 95%CI: 2.1–5.2) and moderates had twice the odds (aOR 2.3; 95%CI: 1.5–3.6) of reporting that the media coverage was the right amount/too much.

The majority (55.2%) responded that the U.S. government had not done enough in response to COVID-19, while 40% felt the government responded correctly. Just 5% felt the government had done too much. Liberals (70.8%) were most likely to respond that the government had not done enough in response to COVID-19, compared to 19.6% of conservatives. Nearly 10% of conservatives reported that the government had done too much in response to the pandemic (p-value <0.001). In the adjusted logistic regression model (Table 2) liberals had 5.7 (95%CI: 3.3–9.7) and moderates had 2.5 (95%CI 1.5–4.3) times the odds of responding that the government had not done enough in response to COVID-19 compared to conservatives. Those who consumed liberal leaning news media and who indicated there was scientific agreement about global warming also had higher odds of feeling the government had not done enough.

Approximately 19% felt that people were generally overreacting to COVID-19 and 40% felt that people were generally under-reacting. Conservatives (36.5%) were most likely to feel that people were overreacting while liberals (44.6%) were most likely to feel that people were under-reacting (p-value <0.001). In the multinomial logistic regression model (Table 2), compared to conservatives, liberals had approximately three times the odds of reporting people were responding correctly (aOR 2.9; 95%CI 1.6–5.3) or under-reacting (aOR: 3.0; 95%CI: 1.6–5.5). Females and those who consumed liberal news had significantly higher odds of feeling people were under-reacting to COVID-19.

Despite variation in opinions regarding the response to COVID-19, 80% felt that their state would experience a major outbreak of the disease. A similar percentage of liberals and moderates felt a major outbreak would occur (82.4%, 79.3%), while a smaller percentage of conservatives (67.6%; p-value = <0.001) agreed. Regardless of political ideology nearly all respondents (95.8%) reported they would self-isolate in the event that they became ill with COVID-19.

Respondents reported moderate agreement with all four statements evaluating fear related to becoming sick or having a family member become sick with COVID-19. Respondents agreed most strongly that they would be scared if an elderly family member contracted COVID-19 (mean: 5.28; SD: 1.21), followed by a young family member (mean: 5.07; SD: 1.34),

**Table 1. Sample demographics: Overall and by political ideology.**

| | | Political Ideology[a] | | | |
|---|---|---|---|---|---|
| | **Total Sample** | **Leans Liberal** | **Moderate** | **Leans Conservative** | |
| | **n = 1,030** | **n = 523** | **n = 359** | **n = 148** | |
| | **n (%)** | **n (%)** | | | **p- value[b]** |
| Age in years (Mean, SD) | 48.8 (18.7) | 48.6 (18.7) | 45.9 (18.3) | 56.7 (17.8) | <0.001 |
| Sex | | | | | |
| Male | 489 (47.5) | 227 (43.4) | 162 (45.1) | 100 (67.6) | <0.001 |
| Female | 541 (52.5) | 296 (56.6) | 197 (54.9) | 48 (32.4) | |
| Race | | | | | |
| White | 505 (49.0) | 234 (44.7) | 160 (44.6) | 111 (75.0) | <0.001 |
| Black | 241 (23.4) | 146 (27.9) | 76 (21.2) | 19 (12.8) | |
| Hispanic | 123 (12.0) | 63 (12.1) | 50 (13.9) | 10 (6.8) | |
| Asian | 127 (12.3) | 61 (11.7) | 61 (17.0) | 5 (3.4) | |
| Other | 34 (3.3) | 19 (3.6) | 12 (3.3) | 3 (2.0) | |
| Marital Status | | | | | |
| Married | 447 (43.4) | 206 (39.4) | 153 (42.6) | 88 (59.5) | 0.001 |
| Never Married | 377 (36.6) | 211 (40.3) | 135 (37.6) | 31 (21.0) | |
| Widowed/Divorced /Separated | 206 (20.0) | 106 (20.3) | 71 (19.8) | 29 (19.6) | |
| Children Under 18 living at home | | | | | |
| Yes | 298 (28.9) | 145 (27.7) | 123 (34.3) | 30 (20.3) | 0.005 |
| Income as a percentage of the Federal Poverty Level (FPL) | | | | | |
| 0–99% of the FPL | 240 (23.3) | 124 (23.7) | 101 (28.1) | 15 (10.1) | 0.002 |
| 100–199% of the FPL | 166 (16.1) | 83 (15.9) | 59 (16.4) | 24 (16.2) | |
| 200–299% of the FPL | 191 (18.5) | 93 (17.8) | 61 (17.0) | 37 (25.0) | |
| 300–399% of the FPL | 121 (11.8) | 62 (11.9) | 33 (9.2) | 26 (17.6) | |
| 400+ % of the FPL | 312 (30.3) | 161 (30.8) | 105 (29.3) | 46 (31.1) | |
| Received government benefits in the last 6 months | | | | | |
| Yes | 249 (24.2) | 122 (23.3) | 107 (29.8) | 20 (13.5) | <0.001 |
| Education | | | | | |
| High School Education or Less | 208 (26.4) | 96 (18.4) | 85 (23.7) | 27 (18.3) | 0.396 |
| Some College or Associates Degree | 135 (17.2) | 189 (35.1) | 135 (37.6) | 54 (36.5) | |
| Bachelor's Degree | 279 (35.5) | 149 (28.5) | 86 (24.0) | 44 (29.7) | |
| Masters or Advanced Degree | 165 (21.0) | 89 (17.0) | 53 (14.8) | 23 (15.5) | |
| Currently Enrolled in School | | | | | |
| Yes | 113 (11.0) | 58 (11.1) | 48 (13.4) | 7 (4.7) | 0.018 |
| Working for pay before COVID-19 Outbreak | | | | | |
| vYes | 566 (55.0) | 283 (54.1) | 213 (59.3) | 70 (47.3) | 0.040 |
| Flu vaccine for current influenza season | | | | | |
| Yes | 537 (52.1) | 265 (50.7) | 185 (51.5) | 87 (58.8) | 0.210 |
| Frequency of Flu Vaccine over the last 5 years | | | | | |
| Annually | 406 (39.4) | 192 (36.7) | 144 (40.1) | 70 (47.3) | 0.640 |
| At least once, but not every year | 301 (29.2) | 158 (30.2) | 104 (29.0) | 39 (26.4) | |
| Never | 323 (31.4) | 173 (33.1) | 111 (30.9) | 39 (26.4) | |

[a]. Political Ideology calculated using the 10-item Political Polarization in the American Public Survey [11]

[b]. p-value derived from a Chi-square test for the difference of proportions

**Table 2. Political and sociodemographic influences on attitudes about media coverage, government action, and community responses to COVID-19.**

| | Covid-19 is receiving... Too little/the right amount media coverage (n = 738)[a] | | The US government has ... Not done enough in response to COVID-19 (n = 569)[b] | | People are generally ... Responding correctly to COVID-19 (n = 420)[c] | | People are generally ... Under-reacting to COVID-19 (n = 411)[c] | |
|---|---|---|---|---|---|---|---|---|
| | n (%) | aOR (95% CI) | n (%) | aOR (95% CI) | n (%) | aOR (95% CI) | n (%) | aOR (95% CI) |
| Political Ideology[d] | | | | | | | | |
| Leans Liberal | 413 (79.0) | 3.3 (2.1–5.2) | 370 (70.8) | 5.7 (3.3–9.7) | 224 (42.8) | 2.9 (1.6–5.3) | 233 (44.6) | 3.0 (1.6–5.5) |
| Moderate | 251 (68.9) | 2.3 (1.5–3.6) | 170 (47.4) | 2.5 (1.5–4.3) | 141 (39.3) | 1.7 (1.0–2.9) | 139 (38.7) | 1.8 (1.0–3.3) |
| Leans Conservative | 74 (50.0) | 1.0 | 29 (19.6) | 1.0 | 55 (37.2) | 1.0 | 39 (26.4) | 1.0 |
| News Source Bias[e] | | | | | | | | |
| Leans Liberal | 345 (75.9) | 1.1 (0.7–1.8) | 320 (70.3) | 2.7 (1.7–4.2) | 184 (40.4) | 1.3 (0.7–2.1) | 204 (44.8) | 1.9 (1.1–3.4) |
| Moderate | 246 (72.6) | 1.0 (0.7–1.6) | 166 (49.0) | 1.3 (0.8–2.0) | 141 (41.6) | 1.1 (0.7–1.8) | 135 (39.8) | 1.5 (0.9–2.5) |
| Leans Conservative | 104 (64.2) | 1.0 | 49 (30.3) | 1.0 | 67 (41.4) | 1.0 | 47 (29.0) | 1.0 |
| Global Warming Question | | | | | | | | |
| Most scientists think global warming is happening. | 545 (74.4) | - - | 476 (65.0) | 2.7 (2.0–3.9) | 305 (41.6) | 1.3 (0.9–2.0) | 313 (42.7) | 1.6 (1.0–2.4) |
| There is a lot of disagreement as to whether global warming is happening. | 192 (64.9) | - - | 92 (31.1) | 1.0 | 115 (38.9) | 1.0 | 97 (32.8) | 1.0 |
| Vaccine Question | | | | | | | | |
| I trust that the government makes the best decisions when it comes to vaccination requirements. | 449 (72.3) | - - | 304 (49.0) | 0.4 (0.3–0.5) | 289 (46.5) | 1.7 (1.2–2.5) | 224 (36.1) | 1.0 (0.7–1.5) |
| I do not trust the government to make decisions about what vaccinations are required. | 288 (70.6) | - - | 264 (64.7) | 1.0 | 131 (32.1) | 1.0 | 186 (45.6) | 1.0 |
| Sex | | | | | | | | |
| Female | 410 (75.8) | 1.5 (1.1–2.0) | 324 (60.0) | 1.3 (1.0–1.7) | 202 (37.3) | 1.3 (0.9–2.0) | 252 (46.6) | 2.2 (1.5–3.2) |
| Male | 328 (67.0) | 1.0 | 245 (50.1) | 1.0 | 218 (44.6) | 1.0 | 159 (32.5) | 1.0 |
| Race | | | | | | | | |
| White | 361 (71.5) | 1.0 | 243 (48.1) | - - | 224 (44.4) | 1.0 | 196 (38.8) | 1.0 |
| Black | 189 (78.4) | 1.1 (0.8–1.7) | 154 (63.9) | - - | 91 (37.8) | 0.7 (0.4–1.2) | 109 (45.2) | 0.8 (0.5–1.3) |
| Hispanic | 84 (68.3) | 0.7 (0.5–1.2) | 68 (55.3) | - - | 45 (36.6) | 0.4 (0.3–0.8) | 47 (38.2) | 0.5 (0.3–0.8) |
| Asian | 81 (63.8) | 0.5 (0.3–0.8) | 84 (66.1) | - - | 47 (37.0) | 0.4 (0.2–0.6) | 48 (37.8) | 0.4 (0.2–0.6) |
| Other | 23 (67.7) | 0.7 (0.3–1.6) | 20 (58.8) | - - | 13 (38.2) | 0.4 (0.2–1.2) | 11 (32.4) | 0.4 (0.2–1.1) |

[a]. Reference group: ...Too much media coverage

[b]. Reference group: ...Done the right amount/done too much in response to COVID-19

[c]. Reference group: ...Overreactive to COVID-19

[d]. Political Ideology calculated using the 10-item Political Polarization in the American Public Survey [11]

[e]. News Score Bias were calculated using the Ad Fontes Media's source evaluation and was averaged for each respondent [16]

a healthy adult family member (mean: 4.93; SD: 1.39) and themselves (mean 4.86: SD: 1.48). Respondents also reported moderate agreement regarding concerns that the country would not have sufficient healthcare providers (mean: 4.84; SD: 1.35) or supplies (mean: 4.78; SD: 1.40) to meet the needs of those infected with COVID-19.

Regarding events surrounding COVID-19, a majority (64.9%) agreed they were afraid they may not be able to purchase supplies, food, and/or medication they needed. Similarly 64.4% of those who were working before the pandemic agreed they were afraid they may not be able to financially provide for themselves or their families if asked not to work due to social distancing, and 66.8% agreed that they were afraid they would not be able to provide for themselves or their families if they became sick with COVID-19. Table 3 shows the distribution of economic anxieties by sociodemographic factors. After adjusting for the other factors in the table, females and those with lower income had higher mean agreement with all three economic anxieties statements as compared to males and those with higher income.

In general, respondents reported changing their behaviors consistent with public health guidelines for social distancing. Table 4 shows the distribution of changes in behavior. The degree to which people reported their behavior changed differed by political ideology. Liberals were more likely to report a change in their behavior in the desired direction compared to conservatives.

In answer to a direct question, 33.3% reported that their perceived mental health was worse than before the pandemic, while 15.8% reported their mental health was better. We examined the change in depressive symptoms using the change in PHQ-9 scores. For behavior changes related to in-person contact with family, close friends, and colleagues, as contact decreased, there was a slight, but statistically significant increase in depressive symptoms (Table 4). A similar pattern was seen for frequenting your usual place of work, restaurants and bars, and stores. There was no statistically significant association between reduction in travel or contact with strangers and depressive symptoms.

Overall, respondents indicated highest mean agreement that COVID-19 had interrupted their social life (mean: 4.44; SD: 1.61). Table 5 provides the mean agreement scores and t-test analyses for differences in social life, work life, home life, and mental health interruptions due to COVID-19 overall and across child and work status. Those with children at home indicated higher agreement that the pandemic had interrupted social, work, and home lives and hurt their mental health compared to those who did not have children at home. Those who were working before the pandemic similarly reported higher levels of interruption for social, work, and home life and worse mental health as opposed to those who were not working.

The sample was generally knowledgeable about COVID-19. The vast majority (93.6%) correctly identified that the World Health Organization had declared COVID-19 a global pandemic. Nearly all correctly identified that COVID-19 was spread by respiratory droplets from coughs and sneezes (90.0%), and by touching infected surfaces followed by touching your face (91.9%). A smaller, but still large percentage, (77.7%) correctly identified that at the time the survey was distributed (March 31st, 2020) there was no vaccination for COVID-19. A bivariate analysis showed that general knowledge largely differed by media bias and sociodemographic characteristics (sex, race, poverty level, and education). However, after adjusting for all parameters using logistic regression, education was no longer significant, sex was only significant on two of the four questions, while race and income were significant on three of the four questions (Table 6). While media bias was not significant for most questions, removing it from the model worsened model fit.

Compared to white respondents, Black and Hispanic respondents had 0.4 (95%CI: 0.3–0.5) and 0.3 (95%CI: 0.2–0.5) times the odds of correctly reporting that there was not a vaccine for COVID-19 at the time of the survey (March 31[st], 2020); Black respondents had 0.4 (95%CI:

**Table 3. Economic anxieties related to COVID-19 by demographic variables.**

| | I am afraid that I will not be able to purchase the supplies, food, medication etc. needed to provide for my family due to events related to COVID-19 | I am afraid that I will not be able to financially provide for myself and/or my family if I am asked not to come to work due to social distancing policies/practices | I am afraid that I will not be able to financially provide for myself and/or my family if I become infected with COVID-19 and am unable to work |
|---|---|---|---|
| | Means (SD) | Means (SD) | Means (SD) |
| Total Sample—Mean (SD) | 4.03 (1.68) | 3.96 (1.88) | 4.12 (1.84) |
| Sex | | | |
| Male | 3.82 (1.66) | 3.75 (1.90) | 3.85 (1.91) |
| Female | 4.21 (1.67) | 4.20 (1.83) | 4.42 (1.72) |
| p-value (single predictor model)[a] | <0.001 | 0.005 | <0.001 |
| p-value (full model/lsmean)[b] | 0.006 | 0.172 | 0.017 |
| Federal Poverty Level (FPL) - | | | |
| <100% FPL | 4.28 (1.7) | 4.54 (1.59) | 4.63 (1.6) |
| 100–199% FPL | 4.34 (1.56) | 4.39 (1.69) | 4.66 (1.66) |
| 200–299% FPL | 3.96 (1.64) | 4.04 (1.77) | 4.36 (1.63) |
| 300–399% FPL | 3.72 (1.55) | 3.41 (1.99) | 3.62 (1.91) |
| 400+%FPL | 3.82 (1.75) | 3.56 (2.01) | 3.59 (1.97) |
| p-value (single predictor model)[a] | <0.001 | <0.001 | <0.001 |
| p-value (full model/lsmean)[b] | 0.018 | 0.006 | <0.001 |
| Education Level | | | |
| High School degree or less | 4.16 (1.66) | 4.46 (1.61) | 4.40 (1.73) |
| Some college or associates degree | 4.10 (1.62) | 4.18 (1.75) | 4.52 (1.63) |
| Bachelor's degree | 3.95 (1.68) | 3.71 (2.02) | 3.81 (1.97) |
| Masters or more advanced degree | 3.82 (1.81) | 3.54 (1.96) | 3.64 (1.92) |
| p-value (single predictor model)[a] | 0.168 | <0.001 | <0.001 |
| p-value (full model/lsmean)[b] | 0.919 | 0.111 | 0.076 |
| Race | | | |
| White, non-Hispanic | 3.87 (1.7) | 3.54 (1.97) | 3.71 (1.98) |
| Black, non-Hispanic | 4.12 (1.67) | 4.11 (1.84) | 4.3 (1.74) |
| Hispanic | 4.29 (1.76) | 4.48 (1.59) | 4.72 (1.51) |
| Asian | 4.19 (1.46) | 4.15 (1.74) | 4.1 (1.74) |

(*Continued*)

**Table 3.** (Continued)

| | I am afraid that I will not be able to purchase the supplies, food, medication etc. needed to provide for my family due to events related to COVID-19 | I am afraid that I will not be able to financially provide for myself and/or my family if I am asked not to come to work due to social distancing policies/practices | I am afraid that I will not be able to financially provide for myself and/or my family if I become infected with COVID-19 and am unable to work |
|---|---|---|---|
| | Means (SD) | Means (SD) | Means (SD) |
| Other race, non-Hispanic | 4.12 (1.72) | 3.57 (2.4) | 3.76 (2.39) |
| p-value (single predictor model)[a] | 0.042 | <0.001 | <0.001 |
| p-value (full model/lsmean)[b] | 0.393 | 0.042 | 0.156 |

[a]. p-value derived from single predictor F test comparing group means

[b]. p-value derived from full model F test comparing group means after accounting for all other parameter in the table

0.3–0.7) times the odds of correctly reporting that COVID-19 is primarily transferred through respiratory droplets; and Black, Hispanic, and Asian respondents had 0.3 (95%CI: 0.1–0.5), 0.3 (95%CI: 0.1–0.7), and 0.3 (95%CI: 0.1–0.7) times the odds of correctly reporting that one can contract COVID-19 by touching infected surfaces and then touching one's nose or mouth.

Compared to men, women had 2.3 times the odds (95%CI: 1.4–3.9) of correctly reporting that one can contract COVID-19 by touching infected surfaces and then touching one's nose or mouth. Compared to those whose news bias score leaned conservative, those whose news bias score was moderate or leaned liberal had 2.2 (95%CI: 1.2–3.9) and 2.2 (95%CI: 1.3–3.8) times the odds of correctly reporting that COVID-19 is primarily transferred through respiratory droplets.

Greater than 90% of respondents correctly identified fever, cough, and shortness of breath as symptoms for COVID-19. However, a majority also said that nausea (69.5%), aches (53.0%), and nasal congestion (66.7%) were common symptoms of COVID-19. Likewise, more than 85% of respondents correctly identified hand washing (94.1%), not touching your face (90.4%), avoiding contact with sick persons (88.0%), avoiding large groups (89.8%) and sanitizing surfaces (88.6%) as recommendations from the Centers for Disease Control and Prevention (CDC) to prevent COVID-19. A smaller, but still large percentage identified avoiding eating in restaurants and bars (70.6%) as recommended. A little over half (50.4%) correctly identified wearing a facemask in public as not being an official recommendation of the CDC. This recommendation was released on April 3rd, 2020 (four days after the survey was administered).

In comparing COVID-19 to seasonal influenza, the majority (65.5%) correctly identified COVID-19 as having a higher case-fatality rate. However, 22% felt that seasonal influenza and COVID-19 had similar risk of death and 12% reported that seasonal influenza was more deadly than COVID-19. Those who were politically conservative were more likely (p-value <0.001) to say that the seasonal influenza was more deadly than COVID-19 (25.7%) compared to moderates (10.3%) and liberals (9.9%).

## Discussion

Political ideology was the strongest factor associated with attitudes toward the COVID-19 response. This finding is consistent with research suggesting that as new politicized issues emerge, ideology is predictive of adopting beliefs which are suggested to be consistent with an

**Table 4. Influence of political ideology on behavior changes and the impact on behavior changes on mental health.**

| | Total Sample | Change in PHQ9 Score[b,c] | | Leans Liberal | Political Ideology[a] Moderate | Leans Conservative | p-value[e] |
|---|---|---|---|---|---|---|---|
| | n (%) | Mean (SD) | p-value[d] | | n (%) | | |
| **Please indicate the extent to which you have changed your social contact routines/behaviors in response to COVID-19** | | | | | | | |
| **Have in-person contact with family members who live near me** | | | | | | | |
| Much less than usual | 366 (40.00) | 0.69 (3.89) | 0.001 | 208 (39.77) | 112 (31.28) | 46 (31.08) | <0.001 |
| Somewhat less than usual | 217 (23.72) | 0.19 (3.19) | | 111 (21.22) | 78 (21.79) | 28 (18.92) | |
| About the same as usual | 236 (25.79) | -0.33 (2.82) | | 118 (22.56) | 80 (22.35) | 38 (25.68) | |
| Somewhat more than usual | 54 (5.90) | -0.06 (3) | | 9 (1.72) | 34 (9.5) | 11 (7.43) | |
| Much more than usual | 42 (4.59) | -1.17 (4.49) | | 18 (3.44) | 20 (5.59) | 4 (2.7) | |
| **Have in-person contact with close friends** | | | | | | | |
| Much less than usual | 561 (57.66) | 0.6 (3.7) | 0.001 | 313 (59.85) | 179 (50.14) | 69 (46.62) | <0.001 |
| Somewhat less than usual | 195 (20.04) | -0.09 (2.75) | | 101 (19.31) | 57 (15.97) | 37 (25) | |
| About the same as usual | 143 (14.70) | -0.36 (2.62) | | 60 (11.47) | 59 (16.53) | 24 (16.22) | |
| Somewhat more than usual | 44 (4.52) | -0.39 (4.51) | | 12 (2.29) | 25 (7) | 7 (4.73) | |
| Much more than usual | 30 (3.08) | -1.2 (4.42) | | 11 (2.1) | 15 (4.2) | 4 (2.7) | |
| **Have in-person contact with colleagues and work friends** | | | | | | | |
| Much less than usual | 486 (60.90) | 0.48 (3.82) | <0.001 | 268 (51.34) | 163 (45.53) | 55 (37.16) | <0.001 |
| Somewhat less than usual | 134 (16.79) | -0.09 (2.99) | | 64 (12.26) | 47 (13.13) | 23 (15.54) | |
| About the same as usual | 101 (12.66) | -0.13 (3.17) | | 45 (8.62) | 39 (10.89) | 17 (11.49) | |
| Somewhat more than usual | 39 (4.89) | -1.72 (4.36) | | 10 (1.92) | 24 (6.7) | 5 (3.38) | |
| Much more than usual | 38 (4.76) | -0.87 (4.17) | | 11 (2.11) | 21 (5.87) | 6 (4.05) | |
| **Gone to restaurants and bars** | | | | | | | |
| Much less than usual | 731 (77.60) | 0.31 (3.52) | 0.020 | 399 (76.29) | 227 (63.23) | 105 (70.95) | 0.001 |
| Somewhat less than usual | 90 (9.55) | -0.01 (3) | | 39 (7.46) | 41 (11.42) | 10 (6.76) | |
| About the same as usual | 61 (6.48) | -0.61 (3.01) | | 26 (4.97) | 26 (7.24) | 9 (6.08) | |
| Somewhat more than usual | 27 (2.87) | 1.7 (3.94) | | 7 (1.34) | 15 (4.18) | 5 (3.38) | |
| Much more than usual | 33 (3.50) | -0.88 (3.57) | | 10 (1.91) | 20 (5.57) | 3 (2.03) | |
| **Gone to stores (grocery, retail, etc.)** | | | | | | | |
| Much less than usual | 419 (41.12) | 0.56 (4.06) | 0.010 | 225 (43.02) | 142 (39.55) | 52 (35.14) | 0.003 |
| Somewhat less than usual | 302 (29.64) | 0.02 (2.83) | | 160 (30.59) | 99 (27.58) | 43 (29.05) | |
| About the same as usual | 198 (19.43) | -0.22 (2.4) | | 89 (17.02) | 66 (18.38) | 43 (29.05) | |
| Somewhat more than usual | 53 (5.20) | 0.32 (3.8) | | 25 (4.78) | 21 (5.85) | 7 (4.73) | |
| Much more than usual | 47 (4.61) | -0.77 (3.58) | | 22 (4.21) | 22 (6.13) | 3 (2.03) | |
| **Gone to my place of work** | | | | | | | |
| Much less than usual | 395 (60.68) | 0.36 (4.16) | 0.011 | 214 (40.92) | 137 (38.16) | 44 (29.73) | <0.001 |
| Somewhat less than usual | 88 (13.52) | 0.07 (3.63) | | 42 (8.03) | 35 (9.75) | 11 (7.43) | |
| About the same as usual | 109 (16.74) | -0.42 (3.09) | | 52 (9.94) | 38 (10.58) | 19 (12.84) | |
| Somewhat more than usual | 26 (3.99) | -1.15 (3.16) | | 7 (1.34) | 18 (5.01) | 1 (0.68) | |
| Much more than usual | 33 (5.07) | -1.09 (3.95) | | 10 (1.91) | 17 (4.74) | 6 (4.05) | |
| **Virtually communicated with others (email, phone, videoconference, etc.)** | | | | | | | |
| Much less than usual | 50 (5.20) | -0.16 (5.15) | 0.287 | 21 (4.02) | 25 (6.98) | 4 (2.7) | 0.009 |
| Somewhat less than usual | 34 (3.54) | -0.26 (2.53) | | 16 (3.06) | 10 (2.79) | 8 (5.41) | |
| About the same as usual | 253 (26.33) | -0.15 (2.91) | | 113 (21.61) | 90 (25.14) | 50 (33.78) | |
| Somewhat more than usual | 188 (19.56) | 0.18 (3.11) | | 104 (19.89) | 59 (16.48) | 25 (16.89) | |
| Much more than usual | 436 (45.37) | 0.45 (3.83) | | 237 (45.32) | 152 (42.46) | 47 (31.76) | |
| **Have face-to-face contact with others** | | | | | | | |
| Much less than usual | 600 (59.46) | 0.41 (3.58) | 0.068 | 325 (62.14) | 195 (54.47) | 80 (54.05) | 0.224 |

(*Continued*)

**Table 4.** (Continued)

| | Total Sample | Change in PHQ9 Score[b,c] | | Political Ideology[a] | | | p-value[e] |
|---|---|---|---|---|---|---|---|
| | | | | Leans Liberal | Moderate | Leans Conservative | |
| | n (%) | Mean (SD) | p-value[d] | n (%) | | | |
| Somewhat less than usual | 215 (21.31) | 0.14 (2.89) | | 108 (20.65) | 76 (21.23) | 31 (20.95) | |
| About the same as usual | 110 (10.90) | -0.29 (2.8) | | 48 (9.18) | 40 (11.17) | 22 (14.86) | |
| Somewhat more than usual | 43 (4.26) | -0.77 (3.32) | | 15 (2.87) | 22 (6.15) | 6 (4.05) | |
| Much more than usual | 41 (4.06) | -0.61 (4.92) | | 18 (3.44) | 17 (4.75) | 6 (4.05) | |
| **Have in-person contact with those who live in my home** | | | | | | | |
| Much less than usual | 117 (13.03) | -0.2 (4.19) | 0.620 | 58 (11.09) | 53 (14.8) | 6 (4.05) | <0.001 |
| Somewhat less than usual | 93 (10.36) | 0.25 (3.37) | | 43 (8.22) | 36 (10.06) | 14 (9.46) | |
| About the same as usual | 475 (52.90) | 0.1 (2.99) | | 242 (46.27) | 141 (39.39) | 92 (62.16) | |
| Somewhat more than usual | 85 (9.47) | 0.51 (4.71) | | 42 (8.03) | 32 (8.94) | 11 (7.43) | |
| Much more than usual | 128 (14.25) | 0.37 (3.5) | | 63 (12.05) | 55 (15.36) | 10 (6.76) | |
| **Have in-person contact with strangers** | | | | | | | |
| Much less than usual | 660 (68.75) | 0.35 (3.58) | 0.179 | 365 (69.79) | 211 (58.77) | 84 (56.76) | <0.001 |
| Somewhat less than usual | 124 (12.92) | 0.2 (2.6) | | 63 (12.05) | 36 (10.03) | 25 (16.89) | |
| About the same as usual | 105 (10.94) | -0.13 (3.11) | | 40 (7.65) | 42 (11.7) | 23 (15.54) | |
| Somewhat more than usual | 34 (3.54) | -0.12 (4.28) | | 6 (1.15) | 23 (6.41) | 5 (3.38) | |
| Much more than usual | 37 (3.85) | -1 (4.45) | | 17 (3.25) | 16 (4.46) | 4 (2.7) | |
| **Traveled to another city/state/country** | | | | | | | |
| Much less than usual | 557 (74.07) | 0.43 (3.85) | 0.116 | 312 (59.66) | 172 (47.91) | 73 (49.32) | <0.001 |
| Somewhat less than usual | 72 (9.57) | 0.15 (3.34) | | 26 (4.97) | 37 (10.31) | 9 (6.08) | |
| About the same as usual | 69 (9.18) | -0.32 (2.79) | | 30 (5.74) | 31 (8.64) | 8 (5.41) | |
| Somewhat more than usual | 32 (4.26) | -0.19 (3.79) | | 9 (1.72) | 17 (4.74) | 6 (4.05) | |
| Much more than usual | 22 (2.93) | -1.14 (5.05) | | 9 (1.72) | 11 (3.06) | 2 (1.35) | |

[a]. Political Ideology calculated using the 10-item Political Polarization in the American Public Survey [11]

[b]. Not applicable coded as missing

[c]. Change in PHQ-9 score calculated as after-before such that a positive number indicates an increase in depressive symptoms [17]

[d]. p-value derived from F test

[e]. p-value derived from Chi-square test

**Table 5. Life disruption due to COVID-19: Overall, for those with/without children in the home, and for those working/not working for pay.**

| | Total Sample | Children under 18 living in the home | | | Working for pay before COVID-19 | | |
|---|---|---|---|---|---|---|---|
| | | Yes | No | | Yes | No | |
| | Mean (SD)[a] | Mean (SD) | Mean (SD) | p-value[b] | Mean (SD) | Mean (SD) | p-value[b] |
| Events related to COVID-19 have… | | | | | | | |
| interrupted my social life | 4.44 (1.61) | 4.77 (1.44) | 4.3 (1.66) | <0.001 | 4.66 (1.48) | 4.17 (1.73) | <0.001 |
| interrupted my work life or vocation | 3.70 (2.13) | 4.59 (1.72) | 3.33 (2.18) | <0.001 | 4.65 (1.65) | 2.53 (2.06) | <0.001 |
| interrupted my home life | 3.58 (1.95) | 4.27 (1.74) | 3.29 (1.96) | <0.001 | 3.83 (1.89) | 3.26 (1.98) | <0.001 |
| have hurt my mental health | 2.91 (2.05) | 3.5 (2) | 2.67 (2.02) | <0.001 | 3.31 (2.03) | 2.43 (1.96) | <0.001 |

[a]. Anxiety scores were coded from 0–6 and averaged such that higher scores indicate higher levels of anxiety

[b]. p-value derived from Chi-square test

**Table 6. Sociodemographic and media influences on general knowledge of COVID-19.**

| | The World Health Organization has declared COVID-19 a global pandemic (true) | | There is a currently available vaccine for COVID-19 (false) | | COVID-19 is spread person-to-person through inhalation of respiratory droplets when and infected person coughs or sneezes (true) | | One can contract COVID-19 by touching infected surfaces and then touching your nose or mouth (true) | |
|---|---|---|---|---|---|---|---|---|
| | Frequency | Fully Adjusted | Frequency | Fully Adjusted | Frequency | Fully Adjusted | Frequency | Fully Adjusted |
| | n (%) | aOR (95% CI) | n (%) | aOR (95% CI) | n (%) | aOR (95% CI) | n (%) | aOR (95% CI) |
| News Bias Score[a] | | | | | | | | |
| Leans Liberal | 430 (94.5) | 1.1 (0.5–2.4) | 367 (80.8) | 1.4 (0.9–2.2) | 418 (91.9) | 2.2 (1.3–3.8) | 423 (93.0) | 1.5 (0.8–2.8) |
| Moderate | 315 (92.9) | 0.9 (0.5–2.0) | 260 (76.7) | 1.0 (0.7–1.6) | 312 (92.0) | 2.2 (1.2–3.9) | 312 (92.0) | 1.3 (0.6–2.5) |
| Leans Conservative | 151 (93.2) | 1.0 | 124 (76.5) | 1.0 | 137 (84.6) | 1.0 | 147 (90.7) | 1.0 |
| Sex | | | | | | | | |
| Female | 503 (93.0) | -- | 413 (76.5) | -- | 496 (91.7) | 1.5 (1.0–2.4) | 507 (93.7) | 2.3 (1.4–3.9) |
| Male | 461 (94.3) | -- | 387 (79.1) | -- | 431 (88.1) | 1.0 | 440 (90.0) | 1.0 |
| Race | | | | | | | | |
| White | 479 (94.9) | -- | 426 (84.4) | 1.0 | 462 (91.5) | 1.0 | 482 (95.5) | 1.0 |
| Black | 221 (91.7) | -- | 157 (65.2) | 0.4 (0.3–0.5) | 207 (85.9) | 0.4 (0.3–0.7) | 211 (87.6) | 0.3 (0.1–0.5) |
| Hispanic | 110 (89.4) | -- | 82 (66.7) | 0.3 (0.2–0.5) | 106 (86.2) | 0.5 (0.3–1.0) | 108 (87.8) | 0.3 (0.1–0.7) |
| Asian | 122 (96.1) | -- | 107 (84.3) | 0.8 (0.5–1.4) | 120 (94.5) | 1.4 (0.6–3.3) | 115 (90.6) | 0.3 (0.1–0.7) |
| Other | 32 (94.1) | -- | 28 (84.9) | 1.3 (0.4–3.9) | 32 (94.1) | 2.3 (0.3–17.3) | 31 (91.2) | 0.6 (0.1–2.9) |
| Income | | | | | | | | |
| <100%FPL | 215 (89.6) | 1.0 | 160 (66.7) | 1.00 | 211 (87.9) | -- | 209 (87.1) | 1.0 |
| 100–199% FPL | 158 (95.2) | 2.1 (0.9–5.2) | 135 (81.3) | 2.0 (1.2–3.4) | 148 (89.2) | -- | 159 (95.8) | 3.3 (1.3–8.3) |
| 200–299% FPL | 174 (91.1) | 1.0 (0.5–1.9) | 159 (78.0) | 1.4 (0.9–2.3) | 166 (86.9) | -- | 173 (90.6) | 1.2 (0.6–2.3) |
| 300–399% FPL | 114 (94.2) | 1.6 (0.6–3.8) | 97 (80.2) | 1.6 (0.9–2.9) | 109 (90.1) | -- | 113 (93.4) | 1.8 (0.7–4.3) |
| 400+%FPL | 303 (97.1) | 3.4 (1.5–7.6) | 259 (83.3) | 1.8 (1.2–2.9) | 293 (93.9) | -- | 293 (93.9) | 1.7 (0.9–3.4) |
| Education | | | | | | | | |
| 1 (Less than HS) | 11 (84.6) | -- | 6 (46.2) | -- | 12 (92.3) | -- | 12 (92.3) | -- |
| 2 (HS or GED) | 174 (89.2) | -- | 139 (71.3) | -- | 170 (87.2) | -- | 172 (88.2) | -- |
| 3 (Some college) | 226 (93.0) | -- | 186 (76.5) | -- | 218 (89.7) | -- | 222 (91.4) | -- |
| 4 (Assoc Deg) | 130 (96.3) | -- | 101 (74.8) | -- | 116 (85.9) | -- | 124 (91.9) | -- |
| 5 (Bach Deg) | 262 (93.9) | -- | 235 (84.2) | -- | 253 (90.7) | -- | 258 (92.5) | -- |
| 6 (Mast Deg) | 125 (98.4) | -- | 101 (80.2) | -- | 122 (96.1) | -- | 121 (95.3) | -- |
| 7 (PhD or equiv) | 36 (94.7) | -- | 32 (84.2) | -- | 36 (94.7) | -- | 38 (100.0) | -- |

ideology. [18] Suggestions of beliefs that correspond with ideology may be implied by the deliverer of information (e.g. a conservative or liberal lawmaker) or through language cues in information sources, such as media.

Political ideology was further associated with behavior change surrounding COVID-19. This finding is consistent with the Theory of Planned Behavior [9]. As political ideology was associated with attitudes toward the COVID-19 response, it is reasonable to assume that those with attitudes suggesting government or community over-response to the pandemic would be associated with beliefs that recommended behavior changes were unnecessary.

The U.S. political climate continues to affect individual, organizational, and governmental responses as the pandemic evolves. However, our results suggest that, even early in the pandemic, political ideology played a large role in the attitudes and behaviors adopted by U.S.

adults. The ability of political ideology (and related measurements such as news source bias) to predict an individual's attitudes about and adherence to recommended behaviors in response to a public health crisis raises concerns about the efficacy of existing strategies to manage such crises in this era of extreme politicization. [8] This suggests the necessity of developing politically neutral strategies that facilitate effective communication surrounding public health crises.

Sociodemographic characteristics were associated with pandemic-related economic anxieties (sex, income, and race), attitudes toward the community response (sex and race), and knowledge (race) about COVID-19. Many of these discrepancies point to persistent gender, income, and racial inequality in the U.S. These phenomena are particularly well illustrated when analyzing the disproportionate burden of economic anxieties felt by minority races, lower income individuals, and females. Higher economic anxiety would be expected among those in lower income brackets, as they have a reduced ability to weather income loss or unexpected expenses. It is also unsurprising that racial minorities in the U.S. are experiencing higher economic anxieties, given the conflation of poverty and race in the U.S.

Increased economic anxiety in females is consistent with other research. This may be at least partially explained by poorer perceived economic stability relative to males. [19] Disparities in care-giving responsibilities may also help explain sex differences in economic anxieties. Females generally have more care-giving responsibilities for home, children, and family, as dictated by societal tradition. [20] Responsibility for maintaining family schedules and routines during this pandemic would likely add disproportionately to the physical and emotional strain on females in the U.S.

While general knowledge about COVID-19 was high, most respondents also identified symptoms including nausea, aches, and nasal congestion which were not part of the initial symptom list. This finding may reflect the emerging nature of information about COVID-19 or inaccurate information spreading by word-of-mouth rather than official sources. While general knowledge about COVID-19 was widely exhibited across most sociodemographic and political characteristics (a promising demonstration of the wide reception of public health messages and recommendations), Black and Hispanic respondents were generally less likely to respond correctly to knowledge questions, which may be a result of a larger proportion of Black and Hispanic respondents lacking access to adequate resources or receiving misinformation. This is particularly concerning given racial differences in the rate of severe complications and deaths from COVID-19. [21] At the time of writing, 48% of deaths due to COVID-19 in Chicago had occurred in African Americans, despite the fact that their percent of confirmed cases (33%) mirrored their proportion of the Chicago population (30%). [21] Unfortunately, these racial and economic disparities mirror well-documented disparities for many other respiratory infectious diseases, including severe outcomes from influenza. These disparities, rooted in historic, racially-motivated policies, limit African Americans' access to care and information and exacerbate factors that place them at higher risk for pre-existing conditions.

After only two weeks of social distancing in most areas of the country, one-third of respondents reported worse mental health than before COVID-19. This finding is consistent with research which identifies social isolation as a significant factor in mental health. As social distancing fundamentally requires separation from most sources of community (i.e. work, religious communities, friends, family, etc.), increases in loneliness as the pandemic progresses may be expected. [22] Social isolation and loneliness are linked to significant increases in morbidity and mortality, which raises concerns about population well-being in the event of protracted social distancing and supports the need to find means of social connection that are consistent with social distancing recommendations. [23]

Disruption to social, work, and home life and worsened mental health due to COVID-19 were higher for those with children at home and for those who were working for pay before

the pandemic. Due to school closures, many with children at home are managing new roles as full-time caregivers and managing educational activities, often while maintaining their own employment responsibilities. Higher levels of disruption and worsened mental health among the employed likely results from disruption of daily routines, job insecurity, or an absence of valued social interaction.

## Strengths & limitations

We acknowledge that these results were based off a cross-sectional study regarding an emerging infection. At the time of data collection, information about COVID-19 was nascent. Knowledge, best practices, attitudes, and impacts have rapidly changed since the collection of these data. Therefore, the generalizability of our results are limited to adult populations in the U.S. during the early weeks of the pandemic's influence in the U.S. Nevertheless, early stage information regarding this pandemic, may prove useful for future outbreaks of emerging infections.

This study was conducted approximately two weeks after implementation of initial social distancing guidelines. As such, it provides the opportunity to examine the early impacts of COVID-19 and associated social distancing in the U.S. population. Although this provides useful information, it is unlikely to represent the attitudes, anxieties, and behaviors of the population throughout the pandemic. Quota sampling for sex, race, and income provided a sample that is statistically similar to the overall population of U.S. adults; however, samples derived from internet panels may differ in unmeasurable ways from the U.S. population. Our sample under-represents households with children at home (28.9% of sample vs. 45.0% U.S. households). [10]

## Conclusion

These findings underscore the need to develop public health messaging that considers the influence of the political climate. Strict fact-based messaging may simply be insufficient to engage the community in desired public health actions, particularly for highly politicized events such as COVID-19. Public health experts should consider differential messaging that appeals to the values of those across the political spectrum.

## Acknowledgments

We thank William F. Christensen for his comments and feedback.

## Author Contributions

**Conceptualization:** Sarah R. Christensen, Emily B. Pilling, J. B. Eyring, Grace Dickerson, Chantel D. Sloan, Brianna M. Magnusson.

**Formal analysis:** Sarah R. Christensen, Brianna M. Magnusson.

**Methodology:** Sarah R. Christensen, Chantel D. Sloan, Brianna M. Magnusson.

**Resources:** Chantel D. Sloan, Brianna M. Magnusson.

**Supervision:** Chantel D. Sloan, Brianna M. Magnusson.

**Writing – original draft:** Sarah R. Christensen, Emily B. Pilling, J. B. Eyring, Grace Dickerson, Chantel D. Sloan, Brianna M. Magnusson.

**Writing – review & editing:** Sarah R. Christensen, Emily B. Pilling, J. B. Eyring, Grace Dickerson, Chantel D. Sloan, Brianna M. Magnusson.

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
