## [Decision Letter · Decision Letter 0]

29 Jul 2020

PONE-D-20-15888

Political and personal reactions to COVID-19 during initial weeks of social distancing in the United States

PLOS ONE

Dear Dr. Magnusson,

Thank you for submitting your manuscript to PLOS ONE. After careful consideration, we feel that it has merit but does not fully meet PLOS ONE’s publication criteria as it currently stands. Therefore, we invite you to submit a revised version of the manuscript that addresses the points raised during the review process.

We look forward to receiving your revised manuscript.

Kind regards,

Geilson Lima Santana, M.D., Ph.D.

Academic Editor

PLOS ONE

Additional Editor Comments:

Dear author, thank you for your submission about such an important issue.

However, I believe some work needs to be done in order to improve it.

I suggest structuring your paper according to the Strobe Guideline for cross-sectional studies: https://www.strobe-statement.org/index.php?id=available-checklists

I've also noticied you haven't numbered all pages and lines throughout the paper. PlosOne guidelines asks to include page numbers and line numbers in the manuscript file. Use continuous line numbers (do not restart the numbering on each page). https://journals.plos.org/plosone/s/submission-guidelines

Please, pay especial attention to reviewers comments and suggestions.

Journal Requirements:

2. We note that you have indicated that proceeding to complete the survey was taken as implied consent in your Ethics statement. However, the manuscript indicates that informed consent was provided prior to the survey. Please correct and clarify. If implied consent please indicate whether this consent method was reviewed and approved by your Institutional Review Board.

3. Please note that PLOS ONE uses a single-blind peer review procedure. We would therefore be grateful if you could include in the information that has been redacted for peer review in the manuscript.

Reviewers' comments:

Reviewer's Responses to Questions

**Comments to the Author**

1. Is the manuscript technically sound, and do the data support the conclusions?

Reviewer #1: Yes

Reviewer #2: No

2. Has the statistical analysis been performed appropriately and rigorously? 

Reviewer #1: Yes

Reviewer #2: No

3. Have the authors made all data underlying the findings in their manuscript fully available?

Reviewer #1: Yes

Reviewer #2: Yes

4. Is the manuscript presented in an intelligible fashion and written in standard English?

Reviewer #1: Yes

Reviewer #2: Yes

5. Review Comments to the Author

Reviewer #1: Comments to authors

In this article, authors investigated perceptions, behaviors and impacts surrounding COVID-19 early in the pandemic response in the U.S. general population. A web-based survey was administered to 1030 U.S. adults. The survey assessed political ideology, scientific trust, and media consumption, as well as demographic information, mental health, attitudes, anxieties, impacts, and knowledge related to COVID-19. Results showed that factors such as political ideology played a large role in the attitudes and behaviors adopted by U.S. adults early in the pandemic. Further, one-third of respondents reported worse mental health than before COVID-19.

The study is well designed. The methods are adequately described and the results are clearly presented.

The manuscript has the potential to provide a beneficial addition to the current research concerning the impact of the COVID-19 global emergency. However, there is one main suggestion below.

- There are some key citations left out. Since the study also evaluated mental health outcomes such as depressive symptoms, recent literature concerning the impact of the COVID-19 pandemic on mental health should be considered. In the Introduction section, it would be good to provide a brief overview of the few available studies investigating COVID-19-related mental health outcomes.

In this regard, some recent articles:

Qiu, J., Shen, B., Zhao, M., Wang, Z., Xie, B., and Xu, Y. (2020). A nationwide survey of psychological distress among Chinese people in the COVID-19 epidemic: Implications and policy recommendations. Gen. Psychiatry 33, 19–21. doi:10.1136/gpsych-2020-100213.

Rossi R, Socci V, Pacitti F, et al. Mental Health Outcomes Among Frontline and Second-Line Health Care Workers During the Coronavirus Disease 2019 (COVID-19) Pandemic in Italy. JAMA Netw Open. 2020;3(5):e2010185. doi:10.1001/jamanetworkopen.2020.10185

Wang, C., Pan, R., Wan, X., Tan, Y., Xu, L., Ho, C. S., et al. (2020). Immediate Psychological Responses and Associated Factors during the Initial Stage of the 2019 Coronavirus Disease (COVID-19) Epidemic among the General Population in China. Int J Env. Res Public Heal. 17. doi:10.3390/ijerph17051729.

Talevi D, Socci V, Carai M, Carnaghi G, Faleri S, Pacitti F (2020) Mental health outcomes of the COVID-19 pandemics. Gli esiti della salute mentale della pandemia di CoViD-19. Rivista Psichiatria 55(3):137-144 doi: 10.1708/3382.33569.

Lai, J., Ma, S., Wang, Y., Cai, Z., Hu, J., Wei, N., et al. (2020). Factors Associated With Mental Health Outcomes Among Health Care Workers Exposed to Coronavirus Disease 2019. JAMA Netw. open 3, e203976. doi:10.1001/jamanetworkopen.2020.3976.

Reviewer #2: The manuscript lacks a solid theoretical foundation and the exposition of the background is limited, the introductory section is insufficient to justify the need to carry out the investigation.

The methodology section has not been described in detail, nor the characteristics of the sample, nor statistical aspects for the estimation of the sample size, nor are data provided on the validity and reliability of the instruments used for data collection ...

The results are based on descriptive statistics, so the scope of the derived statements is limited.

Finally, the discussion and conclusions do not meet the minimum standards, one of the reasons is the poor theoretical foundation in the introduction, and this does not allow a good discussion of the data.

6. PLOS authors have the option to publish the peer review history of their article (what does this mean?). If published, this will include your full peer review and any attached files.

Reviewer #1: No

Reviewer #2: No

---

## [Author Response · Author response to Decision Letter 0]

24 Aug 2020

A response to reviewer document has been uploaded.

---

## [Decision Letter · Decision Letter 1]

14 Sep 2020

Political and personal reactions to COVID-19 during initial weeks of social distancing in the United States

PONE-D-20-15888R1

Dear Dr. Magnusson,

We’re pleased to inform you that your manuscript has been judged scientifically suitable for publication and will be formally accepted for publication once it meets all outstanding technical requirements.

Kind regards,

Geilson Lima Santana, M.D., Ph.D.

Academic Editor

PLOS ONE

Additional Editor Comments (optional):

Reviewers' comments:

Reviewer's Responses to Questions

**Comments to the Author**

1. If the authors have adequately addressed your comments raised in a previous round of review and you feel that this manuscript is now acceptable for publication, you may indicate that here to bypass the “Comments to the Author” section, enter your conflict of interest statement in the “Confidential to Editor” section, and submit your "Accept" recommendation.

Reviewer #2: All comments have been addressed

2. Is the manuscript technically sound, and do the data support the conclusions?

Reviewer #2: (No Response)

3. Has the statistical analysis been performed appropriately and rigorously? 

Reviewer #2: (No Response)

4. Have the authors made all data underlying the findings in their manuscript fully available?

Reviewer #2: (No Response)

5. Is the manuscript presented in an intelligible fashion and written in standard English?

Reviewer #2: (No Response)

6. Review Comments to the Author

Reviewer #2: (No Response)

7. PLOS authors have the option to publish the peer review history of their article (what does this mean?). If published, this will include your full peer review and any attached files.

Reviewer #2: No

---

## [Editor Report · Acceptance letter]

17 Sep 2020

PONE-D-20-15888R1

Political and personal reactions to COVID-19 during initial weeks of social distancing in the United States

Dear Dr. Magnusson:

I'm pleased to inform you that your manuscript has been deemed suitable for publication in PLOS ONE. Congratulations! Your manuscript is now with our production department.

Kind regards,

on behalf of

Dr. Geilson Lima Santana 

Academic Editor

PLOS ONE